# The Normative World of Memes: Political Communication Strategies in the United States and Ecuador

Marco López-Paredes *[iD] and Andrea Carrillo-Andrade

Communication Observatory (OdeCom), Pontificia Universidad Católica del Ecuador, Quito 170143, Ecuador; acarrillo745@puce.edu.ec
* Correspondence: mvlopez@puce.edu.ec

**Abstract:** The media convergence model presents an environment in which everyone produces information without intermediates or filters. A subsequent insight shows that users (prosumers) —gathered in networked communities—also shape messages' flow. Social media play a substantial role. This information is loaded with public values and ideologies that shape a normative world: social media has become a fundamental platform where users interact and promote public values. Memetics facilitates this phenomenon. Memes have three main characteristics: (1) Diffuse at the micro-level but shape the macrostructure of society; (2) Are based on popular culture; (3) Travel through competition and selection. In this context, this paper *examines how citizens from Ecuador and the United States reappropriate memes during a public discussion?* The investigation is based on multimodal analysis and compares the most popular memes among the United States and Ecuador produced during the candidate debate (Trump vs. Biden [2020] and Lasso vs. Arauz [2021]). The findings suggest that, during a public discussion, it is common to use humor based on popular culture to question authority. Furthermore, a message becomes a meme when it evidences the gap between reality and expectations (normativity). Normativity depends on the context: Americans complain about the expectations of a debate; Ecuadorians, about discourtesy and violence.

**Keywords:** political communication; memes; social networks





## 1. Introduction

In 1960, the first nationally televised presidential debate occurred: Richard Nixon debated against John F. Kennedy. Since then, political debates have become a turning point for the democratic process. Nowadays, 85 countries (Barberi and Reina 2020) organize them. For instance, in 2020, 73 million people (25% of the Americans) watched the first debate between Donald Trump and Joe Biden (Stelter 2020). On the other side of the world, in Ecuador, since 2021, the Electoral Law obliges candidates to the Presidency to debate. Concepts such as *Homo videns* (Sartori 1998), *audience democracy* (Manin 1997), or *tele-democracy* (Arterton 1987) describe the new mediated politics.

In the era of Web 2.0 and the hipermediations (Scolari 2008), presidential debates are commented by everybody and anybody, without intermediates or filters in diverse platforms simultaneously. The preferred platform is the microblogging site Twitter: It resembles more traditional news media; besides, it is conceived to distribute and share news (Boehmer and Tandoc 2015). For instance, "the 2012 U.S. presidential election has been described as the "Twitter election" (McKinney et al. 2013, p. 556) based on the fact that "Twitter political activity increased dramatically over the course of the campaign" (Houston et al. 2013, p. 549). "Social media are platforms which can bring about the further personalization of politics (...) in how we discuss and document our experience of political issues" (Highfield 2016). One way in which this personalization occurs is through memes. Political memes are units of content that travel through competition and selection and diffuse at the micro-level but shape the macrostructure of society; also, they use and have become cultural items that people can potentially imitate.

This investigation explores how memetics has changed the way to communicate; it addresses memetics as a global phenomenon. The research question that guides this document is *how citizens from Ecuador and the United States reappropriate memes during a public discussion*. The investigation is based on multimodal analysis and compares the most popular memes among the United States and Ecuador produced during the candidate debate (Trump vs. Biden [2020] and Lasso vs. Arauz [2021]).

Citizens express their opinion through memes: "The use of memes tests the horizontal nature of social networks and the relationship between citizens and political actors" (Troi and Chávez 2018). Furthermore, as they are primarily humorous, they often question the authority, the symbolic power. Memes are helpful to guide in the meaning of an event that concerns most citizens: Generally, memes start from this normative (Shifman 2013a) world: "what must be", and then, they recreate the reality. The distance between both scenarios is where humor is developed. However, the criticisms or exaltations also come off from that gap; in other words, memes create humor from a public situation, but one-by-one, they make a parallel normative world shared less obviously. According to (Phillips and Milner 2017), "such expression can inspire divergent responses in divergent audiences (. . .) they are ambivalent. Simultaneously antagonistic and social, creative, and disruptive, humorous and barbed (. . .) to be essentialized as this as opposed to that".

Moreover, a meme, to become one, can be replicated. The key is to use humor, cultural empathy, or social awareness about political issues that affect a specific group or society (Troi and Chávez 2018). In this sense, memes and political debates are capable of framing political issues: "a frame is a thought organizer, highlighting certain events and facts as important and rendering others invisible" (Ryan and Gamson 2006, p. 13). That is why, during the debates, politicians are called to develop winning messages. What is more, according to Birdsell (2017), viewers learn from debates and that they can also influence the formation of opinions about candidates. However, both phenomena depend to some extent on prior knowledge and partisanship.

According to Shifman (2013a), political memes can be classified into (1) persuasive memes, (2) grassroots action memes, and (3) public discussion memes. This last category is the focus of this research. Its units of analysis are memes produced during the candidate debate for presidential elections in Ecuador and the United States. It is essential to consider that, on the one hand, candidate debates in the United States are marquee events. As they are compulsory, participants see them as the opportunity to highlight candidate strengths, "opponents' weaknesses, and avoiding the kind of gaffe that can torpedo message strategies and sink campaigns" (Birdsell 2017, p. 165). Furthermore, the event sets the media agenda and can be decisory for the voting.

The debate was between Joe Biden (Democratic Party) and Donald Trump (Republican Party).

On the other hand, candidate debates in Ecuador are different; the debate held in 2021 between Andrés Arauz and Guillermo Lasso was the first in 34 years. It occurred because of the reforms to the Electoral Law. The political system in Ecuador allows having as many candidates as political movements that can meet specific requirements. For this reason, for 2021, 16 candidates ran for president; however, after the first round of elections, two candidates remain. They were Guillermo Lasso with a conservative ideology, part of the alliance CREO Movement-Social Christian Party, and Andrés Arauz with a more progressist ideology and member of the Union for Hope Movement, created by the former president Rafael Correa and his allies.

## 2. Literature Review

Political activity in social platforms is related to the media convergence model developed by Jenkins et al. (2006) and his further work, "Spreadable media" (Jenkins et al. 2013). According to the first one, involvement is easy to reach as people follow and generate content around individual interests (Navas 2021). Jenkins warns that media convergence implies a modification in the participatory culture as the uses of the communication change;

the scholar describes it as "almost frenetic" (Pérez Tornero 2008). Moreover, in "Spreadable media" (Jenkins et al. 2013), the authors underline that participatory culture is built on networked communities; they do not only shape the message but the way it flows. The members of this community (prosumers) believe their contributions matter and feel some degree of social connection (Jenkins et al. 2006). In this new context, memetics is born.

The evolutive biologist Richard Dawkins (Dawkins 1989) was the first scholar that used the word "meme" to define a "living structure" that propagates by imitation. Dawkins understood a meme as a replicator, and he related it to the evolution theory. The author identified three critical elements of a successful meme: copy-fidelity, fecundity, and longevity: "copy-fidelity is the ability to replicate accurately; fecundity is its speed of replication; and longevity its stability over time" (Marwick 2013, p. 12).

Dawkins' conceptualization has been adapted to communication studies. Memes are comparable to dialects: "Just as P. F. Jenkins describes the birth of a new "dialect" in birds, by "change of pitch of a note, repetition of a note, the elision of notes and the combination of parts of other existing songs", memes born as a cultural mutation" (Dawkins 1989, p. 246). However, even if this approach laid the foundation for the study of memetics, it shows some limitations. For instance, memes reveal interrelationships where "individual expression intertwine with cultural precedents. This mediated intertwine connects participants spread across great distance" (Milner 2016). Then, the categories mentioned above do not consider the subsequent uses and appropriations for memes.

Limor Shifman defines memes as "cultural information that passes along from person to person, yet gradually scales into a shared social phenomenon" (Shifman 2013b, p. 365). Johnson (Johnson 2007) broadens the assertion and determines that memes "can scale into a social phenomenon and trivial elements of popular culture" (p. 27). Lopez-Paredes and Oñate state that "memes from the Internet" consist of the deliberate association of elements (image and text) in the same significant unit. They represent an idea, concept, opinion, or situation (López-Paredes and Oñate 2020). Then, memes permit the registration or catching of critical ideas that are developed in the collective imaginary. Messages become memes when they are: sticky for the unified experience and spreadable so that they can be diversified (Jenkins et al. 2013).

As it has been stated in the lines above, the characteristics of memes can be summarized in: first, "memes diffuse at the micro level but shape the macrostructure of society; they reproduce by various means of imitation" (Shifman 2013b, p. 365). According to Berrocal, Campos-Dominguez, and Redondo (Berrocal et al. 2014, p. 71), this can be read as a limitation for message creation and topics. Prosumers "tend to replay the same message over and over again–the same content but in different versions and formats–, with scarcely any evidence of participative actions as prosumer".

Second, messages become a meme depending on how it travels and reproduces through competition and selection. Appropriation, reappropriation, imitation, and sharing are critical. They, somehow, follow the rules of competitive selection based on intertwining collective contributions (Milner 2016). For this, it is necessary to generate a "personalized expression via social sharing" (Bennett and Segerberg 2012, p. 739). Bennet and Segerberg relate this attribute to the Logic of Connective Action and politics: "The Logic of Connective Action explains the rise of a personalized digitally networked politics in which diverse individuals address the common problems" (Bennett and Segerberg 2012, p. 739). Then, established ideas interact with new, foreign ones.

We have referred to the creators and replicators of memes as "prosumers" and "users", however, if we conceive them as citizens, we can broaden the implications of memes in political life. Tay (2012) outlined that citizen-made political humor memes are an example of engaged citizen discourse. In this context, even we cannot state that this is a new phenomenon (it is comparable to political cartoons), memes allow amateurs and professionals to create and spread a meaning by common language uses.

Shifman (2013a, p. 120) clarifies that: "While some political memes are framed in a humorous manner, others are deadly serious. But regardless of their emotional keying, political memes are about making a point—participating in a normative debate about how the world should look and the best way to get there". This definition is fundamental as the author stresses that memes do not frame just the reality but also create a standardized world; then, the critics and subsequent humor emerge from the gap between what it must be and what it is. Shifman (2013a) creates a taxonomy to classify them: (1) persuasive memes, (2) grassroots action memes, and (3) public discussion memes. Persuasive memes can be understood as the ones created with an obvious intention to support a candidate. They use reason and emotional aspects and ethical, moral, and ideological appeal to do so. Grassroots action memes are the ones that are related to collective action and networks curated or catalyzed by organizations. Public discussion memes use commonplaces and cultural products. In addition, they make a joke about political characters (Chagas et al. 2019). (Milner 2016) underlines that nowadays, memes have to be seen as part of pop culture. Then, as (Foucault 1972, p. 49) states, popular cultural texts discursively construct the objects about which speak. Then, "popular culture not only reflects, but also constitutes world politics" (Weldes and Rowley 2015, p. 19).

*Ecuador and the United States: Political and Cultural Common Aspects*

Ecuador belongs to the third wave of democratization (Huntington 1992); the United States established liberal democracy in Latin America due to the Cold War. Nowadays, in liberal democracies, political events, such as elections, are crossed by the Internet; on one side, it is widespread that citizens use it to express their opinion or generate data—in theory—without fearing repression. On the other side, experience has shown that in this kind of democracies, it is easy to generate polarization among citizens in online discourses (Tkacheva et al. 2013).

Moreover, in the 20th Century, Latinamerica was strongly influenced by American audiovisual production. Purcell (2009, p. 46) describes the phenomenon for Chile. Still, it can easily represent the reality for the region: "Hollywood became a commodity irresistible, which came to be valued by society Chilean without major social distinctions. Hollywood generated a huge social and cultural impact that positioned the United States as a new benchmark of modernity "North American style". Then, humor-sharing among nations is possible because it tends to "relate to the symbols and stereotypes specific to the place and time of their creation, reflecting norms and aspirations, power structures, and collective fears" (Billig 2005, p. 200).

Summarizing the ideas dropped lines above, the "user-generated" globalization is a world phenomenon. Shifman et al. (2014) state that the Internet does provide not only "equal" technological facilities but also complex nexus of practices and choices among users. Humor is one of these practices: e.g., Internet jokes serve as "powerful (albeit often invisible) agents of globalization".

## 3. Materials and Methods

The research question that guides this investigation is: *how citizens from Ecuador and the United States reappropriate memes during a public discussion?* Exploratory (qualitative) research was developed to answer it. A Multimodal analysis supports it. As memetics express in different modes of communication, they are unique for their multimodality (Milner 2016). Furthermore, multimodality is one of the main approaches that interests the Observatory of Communication (OdeCom) from the Pontifical University of Ecuador (PUCE), to which the authors belong.

Shifman (2013a) underlines the need to evaluate memes not as isolated content units but as a semantic set. These interactions are essential to understanding their meanings. Above all, because memes are ephemeral, so their value is based on their context. That is why memes in this study are based on Multimodal analysis.

Memes should be understood as a public conversation with intertextual connections. Memetics implies "aggregate texts" (Milner 2016). Then, every message will be understood by the use of sources that contain the interaction and integration of two or more semiotic resources—text, static images, or moving images (gifs, videos)—or 'modes' of communication (O'Halloran and Smith 2014; Jewitt et al. 2016). The context in which they have been generated is a fundamental part of the analysis.

"In social science, exploration is a broad-ranging, purposive, systematic, prearranged undertaking designed to maximize the discovery of generalizations leading to description and understanding of an area of social or psychological life" (Stebbins 2014, p. 105). The exploratory research is based on examining the characteristics of memes in general and political memes specifically. Table 1 summarizes the main points:

**Table 1.** Main characteristics of memes and how to analyze them.

| Characteristic | Analysis for Political Memes | Definition |
|---|---|---|
| 1. Diffuse at the micro level but shape the macrostructure of society | Persuasive memes, grassroots action memes; and public discussion memes | Persuasive memes: Propositional rhetoric or pragmatic appeal; Seducing or threatening rhetoric or emotional appeal; Ethical and moral rhetoric or ideological appeal; Critical rhetoric or appeal to the source's credibility. Grassroots action memes: Dynamics of collective or hybrid connective action and networks curated by organizations. Public discussion memes: Political commonplace; Literary or cultural allusions; Jokes about political characters; Situational jokes. |
| 2. Cultural items that people can potentially imitate | Content, form, and stance | Content: referencing both the ideas and the ideologies conveyed by it. Form: physical message, perceived through our senses. Stance: when re-creating a text, users can decide to imitate a specific position that they find appealing or use an utterly different discursive orientation. |
| 3. Travel through competition and selection | Scope | Sticky (unified experience) and spreadable (diversified) mentality |

Source: Chagas et al. (2019), Shifman (2013b) and Jenkins et al. (2013).

*Units of Analysis*

This study starts from the basis of the globalization of humor according to the media convergence model. Public discussion memes are an exciting source because they track the stance and scope, essential to understanding reappropriation over imitation. Furthermore, selecting the candidates' debate (Trump vs. Biden on 30 September 2020, and Lasso vs. Arauz on 21 March 2021) refers to the need for a massive amount of information to understand the context and the underlying humor meanings.

The units of analysis were selected on April 2021, using the search engine Google. It is suitable because of its tracker technology that respects user-generated content. It uses the "software crawler, which automatically visits public web pages and follows the links they contain (…). Trackers go from one page to another and store information they find" (Google 2021). Furthermore, Google is the most used search engine in the United States (Marketing Web Consulting 2020) and in Latin America (90.5% of users) (SEO Quito 2016). The use of search engines secures an enormous scope of the memes, which is a fundamental step for their reappropriation.

The search equations were: (1) debate Lasso Arauz + meme and (2) debate Biden Trump + meme. The results were filtered according to the number of interactions and SEO position. The selected memes were traced to the "original" tweet, and after using the snowball technique, the memes with the most activity (engagement rates) became the memes here examined. This research looks into four memes in total, two for each debate.

## 4. Results

### 4.1. Let's Try to Be Serious: The First American Debate

On 30 September 2020, the first debate between Donald Trump and Joe Biden took place. The aim was to present the policies of both candidates around the topics: Supreme Court, fighting COVID-19, economy, racism, and climate change. However, "hardly a minute went by in the 90-min brawl without one of the candidates angrily interrupting the other" (Breuninger and Wilkie 2020). In general terms, candidates tend to attack the rival for every question instead of discussing their plans or policies. Assertions such as "Everything he says is a lie", "you are the worst President we have ever had" were the words that Joe Biden, candidate for the Democratic Party, said to Trump. Trump, the candidate for the Republican party, replied: "Did you use the word smart? Don't ever use the word smart with me because there is nothing smart about you". He declared that the election process would be a fraud because "they cheat". This kind of behavior and speech is common in polarized elections (Cervi and Carrillo-Andrade 2019).

This behavior is classic in politics and is related to the ad hominem argument—which consists of disqualifying the opponent by denigrating the person rather than the idea. It is one of the stages of disinformation, ubiquitous in post-truth politics (Cervi and Carrillo-Andrade 2019). In this way, politicians try to delegitimize their opponent with references "to incompetence, corruption, and that they seek to denigrate the person or the family" (Loaiza 2019).

In this context, citizens described the debate as "a chaos" (Brown 2020) and used social media to give their opinion; in other words, social media began to be full of user-generated content that framed the political event. Table 2 shows two of the most popular memes and interactions developed towards the candidate debate.

**Table 2.** Memes and interactions around the debate Trump vs. Biden.

| Meme 1 | Meme 2 |
|---|---|
| 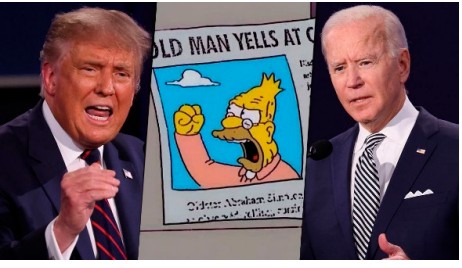 | 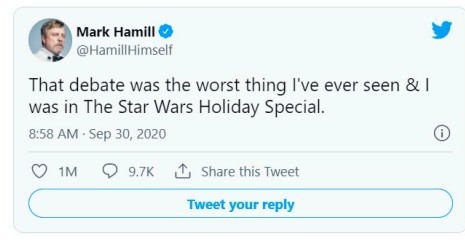 |
| Classification: Public discussion memes | Classification: Public discussion memes |
| Cultural items: Cartoon of grandpa Simpson, under the title "Old man yells at cloud", surrounded by the pictures of Donald Trump and Joe Biden. Critical. | Cultural items: Actor Mark Hamill, who interprets Luke Skywalker in the Star Wars saga; The Star Wars Holiday Special (a spin-off). Critical. |
| Scope: First result under the search engine "debate Trump Biden meme". In this case, 109 interactions from the original meme. | Scope: 1,000,000 likes and 97,000 comments. |

Note: Meme 1 source: Dailymail (2020); meme 2 source: https://twitter.com/HamillHimself/status/1311198722001235968 (accessed on 20 April 2021).

A candidate debate should have been the place to develop persuasive memes as they tend to support candidates and ideologies; instead, these memes are popular because they belong to "public discussion"; they are characterized by the use of commonplaces and cultural products. Meme 1 and 2 represent a "normative" relationship between citizens and public institutions. Parker and Bozeman (2018) affirm that social media plays a role in shaping and responding to shared (public) values. That is why even the political discourse is shaped according to social media trends. Social media has become "a key platform where users converse and promote the application of public values" (Parker and Bozeman 2018, p. 386). In this sense, Meme 1 and 2 frame what the citizens expected, not just about the debate but the candidates. For instance, meme 1 relates Trump and Biden to Grandpa Simpson, a famous character of the well-known show The Simpsons; the grandpa tells long and inaccurate, occasionally inconsistent stories. Then, the use of a grandpa is symbolic as it represents an old patriarch and the values of anger and prepotency, which translates into an old-fashionable speech. A common complaint in U.S. politics is that all presidents, except for Obama, are white, old, wealthy, or well-connected men. In this sense, meme 1 can diffuse at the micro level but shape the macrostructure of society.

The selection of meme 2 was more difficult because even though it respects the conditions of copy-fidelity, fecundity, and longevity (translated into persuasion, identification, and evolution), it seems as it has missing content. It does not use images; nevertheless, López-Paredes and Oñate (2020) remind us that a meme is about creating a narrative rather than combining text and images. Meme 2 calls to the audience's participation; for instance, other celebrities tweeted with this formula. Furthermore, it is trendy to screenshot a tweet and use it as an image (the text becomes an image). Castañeda (2015) argues that a meme obviates any relationship between image and text. Its speech is based on the analog power of the image, which is seen as a whole.

The popularity of the second meme relies on the author rather than on words. Authorship is an important variable to consider concerning scope and spreadability. On the one hand, Boehmer and Tandoc (2015, p. 212), after exploring the reasons for retweeting, conclude that to predict retweeting intentions, "interest in the exact topic of the tweet, the perceived relevance that the tweet might have for the user's followers, and similarity in opinion play important roles". On the other hand, the experiment conducted by Alatas et al. (2019, p. 2) evidences that "endorsements matter: tweets that users can identify as being originated by a celebrity are far more likely to be liked or retweeted by users than similar tweets seen by the same users but without the celebrities' imprimatur". Then, the meme/tweet's popularity relies on the fact that Mark Hamill is famous, with 4.7 million Twitter followers, and because he gives his opinion on a public discussion. The statement is perceived as funny as he remembers The Star Wars Holiday Special, a spin-off of the saga about Life Day, an analogous party to Christmas. The program was aired for the first and only time on 17 November 1978, and was so criticized that no copies were officially released for sale. In this sense, meme 2 also blames the debate as a T.V. show rather than political content. This approach can be more related to the concept of containment. It is about the politics of entertainment with a frivolous, dramatic, and superficial treatment of information. Neither of the candidates presented any data, and the interventions were a mix between opinion and charges.

Mark Hamill was not the only famous that used the mocking technique. Jeremy Slater, the writer of the Fantastic Four, also tweeted: "That was the worst thing I've ever seen, and I wrote Fantastic Four" (https://twitter.com/jerslater/status/1311134678825410560, accessed on 20 April 2021). The same happened with Alec Berg (The Cat in the Hat Movie), the screenwriter Randi Mayem Singer (Tooth Fairy), and others (Know Your Meme 2020). This meme is a good case study about the stance. Even it is supposed to have started with Slater, the most popular tweet was the one of Hamill. Furthermore, it questions the idea of developing "new" memes because they reproduce by various means of imitation (Shifman 2013b).

All in all, the chaotic interventions and the lack of proposals caused a drop of 10 million viewers for the second debate (23 October), which just gathered 63 million. This figure is very far from the 84 million viewers that in 2016, the first debate between Trump and Hillary Clinton garnered.

According to (Branagan 2007), humor plays an essential role in bonding political communities together. "Not only does political humor express a point of view, but it also signals a feeling toward political subject matter" (Penney 2020, p. 792).

### 4.2. Andrés, Don't Lie Again: The Ecuadorian Debate

The debate was broadcast on March 21 and had the aim of helping citizens in identifying the politicians in their actions and proposals. It had an educational format, with a clear division of the sections (economy and job; heath, democracy, and state institutionality; education and technology; international relationships, human mobility, and sustainable development). Every area had an introductory video and infographics to understand the Ecuadorian context before the candidates replied.

In general terms, citizens criticized the form of the debate and the interventions as any candidate went deep enough to clarify how they would deliver on their promises (Table 3). What is more, Arauz insisted on remarking that Lasso is a banker as a synonym of distrust. Lasso repeatedly pronounced the catchy phrase: "Andrés, no mientas otra vez" (Andrés, don't lie again) related to his relationship with the former president Correa. Interestingly, both candidates used data and sources to reply to the questions. Nevertheless, it can be dangerous because the evidence presented was tendentious, contributing to disinformation (Cervi and Carrillo-Andrade 2019), a critical aspect of Ecuador's political system.

**Table 3.** Memes and interactions around the debate Lasso vs. Arauz.

| Meme 3 | Meme 4 |
| --- | --- |
| 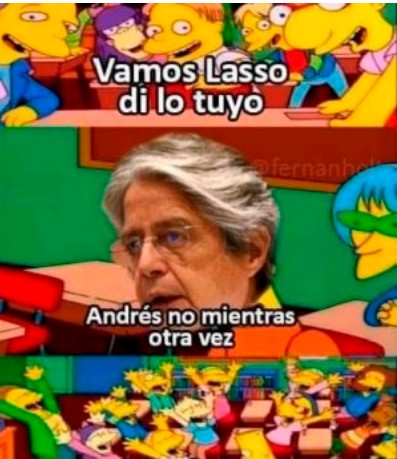 | 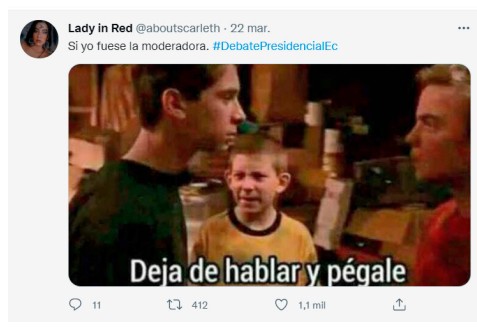 |
| *Copy: "Come on, Lasso, say the line"; "Andrés, don't lie again".* <br> Classification: Public discussion memes <br> Cultural items: Frame series of a famous chapter of The Simpsons with Lasso's catchy phrase against Arauz. <br> Scope: First result under the search engine "debate Lasso Arauz meme"; part of Meme Generator | *Copy: If I were the moderator #PresidentialDebateEC. "Quit talking and hit him"* <br> Classification: Public discussion memes <br> Cultural items: Frame of the famous series "Malcolm in the Middle" with Dewey (the youngest brother" saying, "Quit talking and hit him". <br> Scope: More than 1000 likes, 412 retweets, and 11 comments. |

Note: Meme 1 source: La Verdad (2021); meme 2 source: https://twitter.com/aboutscarleth/status/1373811553309442053, accessed on 20 April 2021.

Meme 3 is a "new" collaboration towards the meme "Say the line, Bart". The frames refer to an episode of The Simpsons in which Bart becomes famous by repeating a catch-phrase. Then, people began to expect Bart to repeat it. This meme is exploitable as it only requires users to change the phrase. Typically, it is used to mock poor exchange of words. The words used by Lasso became viral. They originated a series of videos, memes, and even a song (available in Spotify) with the chorus "Andrés no mientas otra vez" that numbers the cases of corruption of Correa and non-verifiable data that Arauz presented to run the elections. These kinds of phrases are historically handy in Ecuadorian politics, especially by the Social Christian Party. For instance, in 1984, the candidates León Febres-Cordero and Rodrigo Borja met face to face. By that time, the highlight of that debate was the phrase pronounced by Febres-Cordero to his rival: "Look me in the eye, Dr. Borja"; the same is applied to the slogan "bread, shelter, and employment" also owned by Febres-Cordero.

Meme 4 uses a frame of the popular T.V. show Malcolm in the Middle, aired in Ecuador in 2000. The character pictured is Dewey, the youngest brother, generally associated with suppressed anger and disappointment. In this meme, the user refers to the moderator in the copy and asks her (Claudia Arteaga) to provoke a fight instead of an argument: "Quit talking and hit him". Context explains this. As politics in Ecuador usually are related to violence, there is a lack of culture around debates (seen as an exchange of points of view). In 1990, the Congress of Ecuador was the scenario of a big fight between congress members where even the police had to intervene.

Memes 3 and 4 have in common the call for the use of discourtesy, which, according to (Culpeper 1996, p. 366), plays a leading and not marginal role in televised debates. In the words of González Sanz (2010, p. 846), the statement of Culpeper

> stems from the tolerance of the moderators and the few reactions that are produced to repair their image by the offended speakers (...) The insult, expressed especially by improper structures, thus becomes a configuring feature of journalistic debates with political content, within their consideration as spaces that promote aggressiveness and controversy.

In this sense, meme 3 reflects the popularity of the disqualification phrase used by Lasso, and meme 4 shows that citizens were expecting more insults or violence. In Ecuador, insults are helpful to access the electorate and build the discourse: "when the politician insults, he gets closer to the people, to the popular masses, as a form of identification" (Loaiza 2019). What is more, it serves as "access to popular language". Thus the politician who insults "accesses a part of the electorate that the classic, aristocratic politician cannot access" (Hidalgo in Loaiza 2019).

Finally, the users decided to generate humor by imitation rather than develop a different discursive orientation with these memes. So, Ecuadorians use cultural items from American series for re-creating a text. They find appealing these allusions that are not close to the Ecuadorian reality but part of their culture thanks to these "agents of globalization" (Shifman et al. 2014, p. 727).

## 5. Discussion

Memes mutate and adapt constantly, not just in their content but also in their form. The use of text placed on the interface of Twitter is becoming a more popular form of meme; it is read as an image, and even it looks similar to plain text, it gives crucial information as the source. In this sense, some messages become famous by their creator rather than by their words. Nevertheless, these tend to have a short life because common prosumers cannot reproduce them and acquired the same impact: They may be sticky but not easily spreadable (Jenkins et al. 2013). The life of these memes is even shorter if they belong to the public discussion memes category.

Ecuador and the United States share cultural items and humor; this becomes evident when analyzing public discussion memes. Both countries use Simpsons' frames to generate humor and to criticize reality. This evidences how reappropriation occurs as an American T.V. show can be a common tongue for Latin America. In contrast, the debate of the United

States originated new memes and a template for them; that is the case of "That Debate Was The Worst Thing I've Ever Seen". Ecuadorian popular memes were merely an adaptation of old ones that have already been circulating on the Internet. Then, it is crucial to hold in mind that "humorous texts relate to the symbols and stereotypes specific to the place and time of their creation, reflecting norms and aspirations, power structures, and collective fears" (Shifman et al. 2014, p. 727). So, that a developing country shares the cultural items of a member of the G7 shows an inner relationship of power.

Going back to the definition of Shifman (2013b): "political memes are about making a point—participating in a *normative* debate about how the world should look and the best way to get there", this study may suggest that Ecuadorian citizens may be more frustrated than Americans. The American culture shapes Ecuadorian thoughts. Then, the "normative" world they frame in memes is much more distant from their reality. This analysis has also pointed out that popular memes in the United States reflect a desire for a normative world (memes complain about the candidates and develop their ideas in the debate). At the same time, humor relies on reproducing catchy phrases or incitement to violence in Ecuador. Moreover, social allows prosumers to connect within the virtual world (alter world), giving their opinion on how the real world must operate.

Moreover, normativity in social media comes from sharing public values. What occurs with memes is that they give the impression that while all citizens share these public values, politicians do not: Humor and critics originate. Then, memes reflect rivalry between citizens and politicians and normalize problematic perspectives. Nevertheless, what occurs is that messages—because they need to impact—tend to ambiguity and trivialization of issues or even to the manipulation by political elites (Phillips and Milner 2017). What is more, in Ecuadorian politics, populism has reinforced the perception of politicians as "the others"; then, memes tend to spread more when they call to citizen identification against the ones that govern.

The absence of spaces for deliberation in Ecuador causes citizens to criticize the marketing strategy (such as the slogans); they do not focus on the content of the speech; Ecuadorians also expect to witness acts of violence as it has been widespread in their political history. However, this cannot be generalized because, in the American debate, the insults and the ad hominem argument were much more used than in the Ecuadorian. In other words, the debate of the American candidates generated rejection among citizens because it was chaotic and disinformative. In comparison, the educational debate of Ecuador was criticized for the lack of verbal or physical violence.

In terms of the methods, the research presents some limitations. Even though the sample allows an explicit approximation to how citizens frame the debates and how they prefer to interact during a public discussion, it remains small. The value of this research relies on their qualitative methods. Then, this study demonstrates that political memes are a critical object of analysis because they can show prosumers/citizens' expectations regarding their politicians. Memetics carries the potential for public voice and many voices (polyvocality) (Milner 2016). Nonetheless, the analysis must consider those ambivalent interactions in the digital world that build an alternate and normative world.

**Author Contributions:** Conceptualization, M.L.-P. and A.C.-A.; methodology, M.L.-P.; software, A.C.-A.; validation, M.L.-P. and A.C.-A.; formal analysis, M.L.-P. and A.C.-A.; investigation, M.L.-P. and A.C.-A.; resources, A.C.-A.; data curation, A.C.-A.; writing—original draft preparation, A.C.-A.; writing—review and editing, A.C.-A.; visualization, Andrea Carrillo-Andrade; supervision, M.L.-P.; project administration, M.L.-P.; funding acquisition, M.L.-P. All authors have read and agreed to the published version of the manuscript.

**Funding:** This research received no external funding.

**Institutional Review Board Statement:** Not applicable.

**Informed Consent Statement:** Not applicable.

**Data Availability Statement:** The data are available from the corresponding author upon request.

**Acknowledgments:** Thanks to the Observatory of Communication (OdeCom) of the Pontifical University of Ecuador (PUCE), who supplies the time and materials to develop research towards communication using multimodal analysis.

**Conflicts of Interest:** The authors declare no conflict of interest.

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
