# Peer review of "The Normative World of Memes: Political Communication Strategies in the United States and Ecuador"

_journalmedia, doi:10.3390/journalmedia3010004_

Round 1

Reviewer 1 Report

The premise of this paper is intriguing, given the two political contexts it focuses on. Its sections are presented out of order, which, combined with several awkward phrases and grammatical errors, make following the manuscript difficult. Most importantly, the paper doesn’t make a compelling case for the study’s contributions to theory or practice. The study also misses an opportunity to compare the key takeaways from the two case studies (United States and Ecuador).

Copyediting is needed throughout. Here are some examples, but they are not exhaustive: capitalize “americans.” “Summered” should be “summarized.” “Second, a meme becomes one according to way they travel and reproduce” should be “messages become memes depending on the way they travel and reproduce.” “It is appropriate to stay that they follow the rules of 81 competitive selection" should be “to state.” “Political live” should be “political life.”

While discussing debates on page 1, use more pertinent references about the public’s live tweeting of political events. The example provided, “For instance, 50 % aproximately of social-media users report sharing or reposting news stories while being online (Matsa and Mitchell 2014)" is too generic and should be replaced. Perhaps look into these studies:

- Zheng, Pei, and Saif Shahin. "Live tweeting live debates: How Twitter reflects and refracts the US political climate in a campaign season." Information, Communication & Society 23, no. 3 (2020): 337-357 or

- Houston, J. Brian, Mitchell S. McKinney, Joshua Hawthorne, and Matthew L. Spialek. "Frequency of tweeting during presidential debates: Effect on debate attitudes and knowledge." Communication studies 64, no. 5 (2013): 548-560 or 

- Bramlett, Josh C., Mitchell S. McKinney, and Benjamin R. Warner. "Processing the political: Presidential primary debate “live-tweeting” as information processing." An Unprecedented Election: Media, Communication, and the Electorate in the 2016 Campaign (2018): 169.

Citation and page number are missing for this paragraph: “Dawkins’ conceptualization has been adapted to communication studies. In fact, memes are comparable to dialects: “Just as P. F. Jenkins describes the birth of a new “dialect” in birds, by “change of pitch of a note, repetition of a note, the elision of notes and the combination of parts of other existing songs”, memes born as a cultural mutation”.  All direct quotes require inclusion of page number in the citation.

On page 2, it is stated that “Third, memes travel through competition and selection.” The rules of competitive selection were covered under the second point. The abstract hints that the second point should be about cultural attributes.

The literature review relies too heavily on Shifman references. What have previous mass-communication studies found about the use and sharing of political memes? How does the current study contribute to that body of knowledge?

Overall, the paper is disjointed. Research questions should follow immediately after the literature review and context. The methods section should precede the results. The paper should include a discussion & conclusion section — what are the study’s contributions and limitations?

Author Response

Dear reviewer:

Thank you for your comments and insights. We examined the sources you suggested; we incorporated them in the introduction, especially for the U.S. contextualization. Besides, as you warned, Shifman's literature was not enough to understand political memes. In that sense, we included Milner (2016) and Jenkins, Ford, and Green (2013) to have a variety of voices and ideas and contrast Shifman's assertions.

Also, we modified the conclusions to enlarge our contributions. You may find it interesting how we framed ambiguity and trivialization in memetics.

In terms of form, we sent the document through a copyediting process that included the review of citations and sources. However, we must point out that, as some sources are Kindle Edition, we could not have pages numbers. 

Finally, about the paper being disjointed, we respected the layout proposed by the journal. However, before the results, we included the research question in the introduction so it could be more evident. Moreover, we clarified the methods we used. 

It has been very enriching to hear your comments and review your literature.

Reviewer 2 Report

The core research inquiry regarding normative qualities of political internet memes in Ecuador vs. the U.S. is of value to both the literatures on memes and public conversation regarding politics. However, the manuscript needs a good deal of improvement.
The paper seems to be out of order, with the Methods section appearing last, after the Discussion. The Discussion should follow a clear explanation of the Methods and Materials. The included research question is too broad, and would be better if tailored to reflect the paper's focus on normative discourse appearing in memes and the comparison between the U.S. and Ecuador. More description of the process for collecting and analyzing the memes is needed. Also, what does it mean that multimodality comes from OdeCom? Multimodal Discourse Analysis has been used with internet memes before, such as by Ryan Milner (2012, 2016). There needs to be a clearer description of the analysis process and the results. Right now there seems to be a lot of description of a few examples and it's difficult to understand the broader themes.
Regarding tweets as memes, Shifman (2013) also describes a difference between viral media and memes, in that memes have many iterations and participants. The fact that other celebrities tweeted with this formula is a better defense of the tweet being a meme than the others given.
The paper has many misspellings, direct quotes without page numbers, and other grammatical errors. 
I think if these concerns are addressed, the paper could have value. The question of memes reflecting normative views of politics respective to different countries -- even when U.S. pop culture is used as the vehicle for doing so -- is an interesting one. Likewise, understanding political discourse in Ecuador is of value.

Author Response

Dear reviewer:

Thank you for your comments and insights. Firstly, we would like to share that we have rethought our research question. Now, we framed our document as an examination of how citizens from Ecuador and the United States reappropriate memes during a public discussion. In this sense, we adapted our results and discussion. 

Secondly, about the selection of the method, we clarified it. Also, we added the insights of Milner (2016) for our study, as you suggested. However, we did not mean that multimodality comes from OdeCom. We wanted to state that multimodality is one of OdeComs's primary concerns. In this sense, we are trying to specialize in this approach.

Thirdly, in the document's body, we included your insight about popular culture through the literature of Milner (2016) and Weldes and Rowley (2015). Moreover, we find absorbing your comment on the tweet/meme, so we did further research and supported your affirmation with Jenkins, Ford, and Green (2013). 

Fourthly, we sent the document through a copyediting process that included the review of citations and sources. However, we must point out that, as some sources are Kindle Edition, we could not have pages numbers. 

Finally, about the paper being disjointed, we respected the layout proposed by the journal. However, before the results, we included the research question in the introduction to be more evident. 

It has been very enriching to hear your comments and review your literature.

Reviewer 3 Report

I recommend to revise and resubmit as I think that the article can be better situated in the current discussions of memes and political polarization/social media politics, addressing the concerns below.

While the argument seems to be that memes mutate and adapt to the national context, I missed a conceptual discussion spelling out the exact contribution of the paper to the discussion of memes as engaging objects of political communication. The insight that the political culture of the country is what determines the content of each meme is rather general when not usefully rearticulated through the contexts of everyday politics of social media.  

The author acknowledges the creation of normative narratives through memetic political communication but fails to connect the discussion of national contexts to the discussion of ambiguity that memes reinforce as they spread on social media platforms, claiming that a meme “becomes popular by its creator rather than by its words” instead. Such a claim should be better explained, when not perhaps even questioned, especially since memes tend to disrupt the logic of source and adaptation as they circulate between different national and platform contexts. 

With respect to the various definitions of memetic cultures employed throughout the text, the proposition to situate memes within the discussion of “convergence culture” is by all means ‘safe’.  However, I think that Jenkins et al.'s work on “spreadable media” might be more suitable here, especially since the paper (drawing on Shifman) rightfully highlights the multimodal and collective nature of memes in relation to their capacity to create and spread a meaning.

In this context, some important concepts from the areas of Internet research and media studies could be stronger acknowledged and combined with the author’s ideas: Tim Highfield’s discussion of social media and everyday politics could offer a fruitful framework for developing the idea of political context as mediated through Twitter. With respect to the exploration of political memes and humor see e.g., Joel Penney’s work on memes and the US political expression in Television and New Media. The overall problematization of memes as ambiguous communicative media objects could help to tease out the disruptive potential of Internet humor as mediated through the logic of polarisation/amplification that social media networks set into motion (see e.g the work of Jodi Dean on the communicative capitalism and Phillips & Milner’s explorations of the “ambivalent Internet”). Last but not least, the author might also find interesting the work of Megan Boler and Elisabeth Davis with particular focus on how Twitter and other social media can be used to amplify political conflicts. 

Author Response

Dear reviewer:

Thank you for your insights. It has been very enriching to read your comments and review your literature. After we read the literature suggested by you, we made significant changes to the paper. 

First of all, to address the issue of ambiguity, we relied on Phillips and Milner (2017). In this sense, and for our specific study of Ecuadorian and U.S. politics, we concluded that:

normativity in social media comes from sharing public values. What occurs with memes is that they give the impression that while all citizens share these public values, politicians do not: Humor and critics originate. Then, memes reflect rivalry between citizens and politicians and normalize problematic perspectives. But what occurs is that messages —because they need to impact— tend to ambiguity and trivialization of issues or even to the manipulation by political elites (Phillips and Milner 2017). What is more, in Ecuadorian politics, populism has reinforced the perception of politicians as "the others"; then, memes tend to spread more when they call to citizen identification against the ones that govern.

Secondly, Jenkins, Ford, and Green's (2013) literature was beneficial to enrich our literature review. Also, it let us compare Shifman's assertions. Then, the concepts of "sticky" and "spreadable" were very useful to face the implications of famous people tweeting, which can be an exciting contribution of this paper. 

Thirdly, we included Highfield (2016) in our introduction to understand the role of social media in political communication. 

Fourthly, about the selection of the method, we clarified it. Also, we added the insights of Milner (2016) for our study to support the election of multimodality. 

Finally, we sent the document through a copyediting process that included the review of citations and sources. Then, we are sure that the document is more precise and well-referenced. 

Once again, thank you for sharing this literature and for taking the time to give us your feedback.

Reviewer 4 Report

The authors undertake a comparison of political memes observed in the United States and Ecuador. The literature review is interesting, as is the discussion. I do believe this work would benefit from a clearer presentation of the argument. The meme gathering methodology does not appear to be comprehensive, which is fine so long as there is a justification for the sorts of memes that were observed. Overall, this was enjoyable to read and consider.

Author Response

Dear reviewer:
Thank you for your comments and insights. We worked on your suggestion about the presentation of the argument. In this sense, we added literature around Highfield (2016); McKinney, Houston, and Hawthorne (2013); Jenkins, Ford, and Green (2013); and Phillips and Milner (2017). We were more capable of contextualizing political communication in social media and focusing on the similarities and differences between U.S. and Ecuador. 

We also addressed your concern about the gathering methodology in the section "Units of analysis": 

The selected memes were traced to the "original" tweet, and after using the snowball technique, the memes with the most activity (engagement rates) became the four memes here examinated. 

All in all, we are very grateful for your positive comments and thoughtful insights. 

Round 2

Reviewer 1 Report

This paper has an exciting topic. The research is timely and reveals some interesting findings on memes in different political contexts. The theoretical context has improved from the first version. However, I have a few suggestions:

It appears that the authors, by employing the Word template for journal submissions, are taking the order of suggested sections literally, which significantly hurts the flow of the paper. A quick search of recent published articles in the journal shows that research studies should follow the classic order of introduction > literature review and RQs > methods > results > discussion. Indeed, the instructions for authors specify the following: "We do not have strict formatting requirements, but all manuscripts must contain the required sections: Author Information, Abstract, Keywords, Introduction, Materials & Methods, Results, Conclusions, Figures and Tables with Captions, Funding Information, Author Contributions, Conflict of Interest and other Ethics Statements."

Even if the sections get rearranged, more work is needed within each section. 

The introduction is too long. A statement of objectives is only included on page 4. The key ideas of the research, a brief rationale for the study (why it’s important, what insights it hopes to achieve, what its contributions are) and a summary of how its goals will be accomplished need to be included in the first couple of paragraphs. 

The information from the second paragraph onward need sto be included in a new Literature Review section.

The methods section should specify when the search for each debate was conducted. Immediately after each debate or at a later time? Are search engines the best tool for finding Twitter memes? Also, specify that two tweets per debate were selected (it’s unclear if the corpus of four reflects the entire sample or if it’s four memes per debate).

The first section under Results should be moved under the introduction or literature review. 

The background information on the American and Ecuadorian debates should also move to dedicated sections in the introduction or literature review to set up the stage for the results. The results should start with the line that reads, “Table 1 shows two of the most popular…” How popular was meme 3? The "Scope" doesn’t mention its reach/engagement level, as defined in the methods section.

The Discussion section should include a brief overview of the study’s limitations and a conclusion highlighting key findings and their implications.

Author Response

Dear reviewer,

Thank you very much for your thoughtful feedback. We considered all your observations.

First of all, we changed the sections' order as suggested, according to the Journal parameters. The article has an Introduction, Literature Review, Materials and Methods, Results, and Discussion. Also, the introduction includes the aim of the research, its justification, and the contextualization of the debates. Besides, the state of the art comes in the Literature Review.

Second, the section "Materials and Methods" was improved. We clarified the use of search engines and the time in which the research was held. Also, we specified the sample of our units of analysis. 

Third, as you suggested, in the Conclusion, we added a brief overview of the study's limitations and a conclusion highlighting key findings and implications.

Finally, we resent the document through a copyediting process. Besides, we added the position or number page in the case of Kindle or e-book sources.

It has been very enriching to read your comments.

Reviewer 2 Report

The author(s) may need to clarify with the editor regarding the order of paper sections. As I stated in my previous review, the article does not make sense with the sections in the order presented and it hurts the flow and quality of the paper.

Indeed, the Instructions for Authors regarding Free Format Submission state: “We do not have strict formatting requirements, but all manuscripts must contain the required sections: Author Information, Abstract, Keywords, Introduction, Materials & Methods, Results, Conclusions, Figures and Tables with Captions, Funding Information, Author Contributions, Conflict of Interest and other Ethics Statements.”

The discussion really needs to come last for a research article of this type.

Additionally, it’s still unclear how these examples were selected and analyzed. The idea of a paper comparing normative elements of meme use in response to debates in the U.S. vs Ecuador is still of merit, but the manuscript needs a good deal of work to establish its own rigor in doing so before it can merit publication. Unfortunately, the reworking of the preceding literature seems to have introduced new grammatical errors as well.

Additionally, the author(s)’ comments regarding the inability to cite page numbers with Kindle editions of books raises some concerns regarding rigor. Please review Chicago guidelines here: https://www.chicagomanualofstyle.org/book/ed17/part3/ch14/psec160.html. It is also possible to find page numbers in Kindle: https://www.howtogeek.com/715778/how-to-see-a-books-page-number-on-amazon-kindle/

Author Response

Dear reviewer,

Thank you very much for your thoughtful feedback. We considered all your observations.

First of all, we changed the sections' order as suggested, according to the Journal parameters. The article has an Introduction, Literature Review, Materials and Methods, Results, and Discussion. Also, the introduction includes the aim of the research, its justification, and the contextualization of the debates. Besides, the state of the art comes in the Literature Review.

Second, the section "Materials and Methods" was improved. We clarified the use of search engines and the time in which the research was held. Also, we specified the sample of our units of analysis. 

Third, in the Conclusion, we added a brief overview of the study's limitations and a conclusion highlighting key findings and implications.

Finally, we resent the document through a copyediting process. Besides, we added the position or number page in the case of Kindle or e-book sources.

It has been very enriching to read your comments.

Round 3

Reviewer 1 Report

The revised manuscript flows better, it is more adequately situated within the existing body of literature, and the research design is more transparent. Some grammar issues persist, but they can be addressed during the final copy editing stage.

Reviewer 2 Report

Overall, the paper is much improved with the structural changes. I do think the cross-cultural look at norms is of interest and usefulness to the literature. There are still some editing/grammar issues.